# Single Dense Layer of PEO Coating on Aluminum Fabricated by “Chain-like” Discharges

**DOI:** 10.3390/ma15134635

**Published:** 2022-07-01

**Authors:** Liye Zhu, Wei Zhang, Haitao Liu, Lei Liu, Fuhui Wang, Ziping Qiao

**Affiliations:** 1College of Chemistry and Life Sciences, Chifeng University, Yingbin Road 1, Chifeng 024000, China; lht1803@163.com; 2Inner Mongolia Key Laboratory of Photoelectric Functional Materials, Yingbin Road 1, Chifeng University, Chifeng 024000, China; 3Shi-Changxu Innovation Center for Advanced Materials, Institute of Metal Research, Chinese Academy of Sciences, Wencui Road 62, Shenyang 110016, China; 4Guangdong Provincial Key Laboratory of Modern Surface Engineering Technology, Institute of New Materials, Guangdong Academy of Sciences, Guangzhou 510650, China; 5School of Materials Science and Engineering, Shenyang University of Technology, Shenyang 110870, China; 15848022846@126.com; 6Key Laboratory for Anisotropy and Texture of Materials (MoE), School of Materials Science and Engineering, Northeastern University, Wenhua Road 11, Shenyang 110819, China; fhwang@mail.neu.edu.cn; 7Science and Technology on Transient Impact Laboratory, No. 208 Research Institute of China Ordnance Industries, Beijing 102202, China; 12007010@208mail.com

**Keywords:** “chain-like” discharges, PEO coating, single dense layer, properties

## Abstract

Reducing the loose-layer-to-dense-layer ratio in PEO coatings on aluminum and its alloys is the key to improving their corrosion resistance and expanding their applications in the aerospace industry and other fields. In this paper, we describe the discharge evolution during the PEO process in exhaustive detail and report the appearance of a novel “chain-like” discharge for the first time. We investigated the microstructure and composition of PEO coatings using a scanning electron microscope (SEM) equipped with an energy-dispersive spectrometer (EDS) and an X-ray diffractometer (XRD). The results reflected that the coating composition changed from amorphous Al_2_O_3_ to crystalline γ-Al_2_O_3_ and α-Al_2_O_3_ phases with the evolution of the plasma spark discharge state. We evaluated the electrochemical behavior of the coatings using a potentiodynamic polarization curve and electrochemical impedance spectroscopy (EIS) in 3.5 wt.% NaCl solution. Under “chain-like” discharge, the i*corr* of the coating on Al was 8.564 × 10^–9^ A∙cm^−2^, which was five orders of magnitude lower than that of the sample without the PEO coating. Moreover, we evaluated the adhesion strength of the coatings at different stages using a pull-off test. The adhesion strength of the PEO coatings at stage V reached 70 MPa. Furthermore, the high content of α-Al_2_O_3_ increased the hardness of the coating to 2000 HV. Therefore, the “chain-like” discharge promoted the formation of a single dense layer with 2.8% porosity and that demonstrated excellent properties. We also propose a mechanism to explain the influence of the plasma spark discharge state on the microstructure and composition of the PEO coatings.

## 1. **Introduction**

Aluminum alloys are widely used in the marine, chemical, aerospace, metal packaging, and transportation industries, among others, because of their low density, advantageous mechanical properties, efficient processability, non-toxicity, easy recovery, and excellent conductivity and heat transfer characteristics [1]. The amorphous oxide film with a thickness of 4–5 nm that naturally forms on aluminum and its alloys in the atmosphere protects the internal metal from corrosion, but it does not improve the shortcomings of aluminum alloys, such as their low surface hardness, poor wear resistance, and poor corrosion resistance in harsh environments (including alkaline conditions and marine environments) [2]. Consequently, various surface treatment methods, such as electroless plating, electroplating, physical vapor deposition (PVD), chemical vapor deposition (CVD), and anodic oxidation, are used to improve the surface properties of aluminum and its alloys [3,4,5,6,7]. However, the application of these surface treatment methods is restricted, due to their environmental pollution. In this context, academics and industry professionals have paid great attention to plasma electrolytic oxidation (PEO), which meets the requirements of aluminum alloy components, such as the engines and propeller blades of hovercraft, sonar systems, pistons, spinning cups, and automobile oil pumps, and is also highly effective and environmentally friendly [8,9,10].

A PEO coating has a three-layer structure, i.e., a porous outer layer, dense inner layer, and interface barrier layer. The outer layer, with its relatively loose and porous structure, can easily be powdered during application and increases the surface roughness of the PEO coating, which decreases the wear resistance. Furthermore, a porous layer with many through flaws diminishes the corrosion resistance of the PEO coating. The dense layer is the main body of a PEO coating, accounting for approximately 2/3 of the thickness of the oxide layer. This layer is compact, the diameter of each pore is several microns, and the porosity can be restricted to below 5%. The diameter of the pores and the porosity directly determine the corrosion resistance of the coating. The third layer in a PEO coating is the interface barrier layer, which lies between the oxide layer and the matrix. The barrier layer and the matrix penetrate each other via a typical metallurgical bonding process, significantly enhancing the bonding force of the coating [2]. Therefore, reducing the ratio of the porous layer to the dense layer is key to improving the properties of a PEO coating. Many researchers have attempted to produce a compact coating by altering the electrolyte [11,12], including additives [13,14], and adjusting the electrical parameters [15,16,17]. These techniques increase the density of a coating by changing the discharge state, especially by producing “soft” sparks, which dramatically promotes the growth of the dense layer [18,19]. Evidently, this discharge state plays an important role in the microstructure of a PEO coating.

Over the past few years, many researchers have reported the appearance of “soft” sparks after properly adjusting waveform parameters [20,21]. Unfortunately, little is known about the evolution of discharge after the appearance of soft sparks or their influence on the microstructure of PEO coatings. Although one can produce a thick, homogeneous, and dense layer with a reduced porous-layer-to-dense-layer ratio by changing from the traditional “arc“ discharge method to the “soft” PEO system, one cannot eliminate the porous outer layer [22,23,24,25]. We propose a method for preparing a single dense layer, by controlling the transition from a soft-spark to “chain-like” discharge, and elucidate the evolution of discharge across the whole PEO process. Through a comprehensive analysis of the voltage–time curve, plasma spark discharge states, and morphology changes in the PEO coating, we divided the process into five stages and determine the relationship between the discharge evolution and the microstructure of the PEO coating. Finally, we investigated the influence of the microstructure on the corrosion resistance and mechanical properties of the coating. Aluminum alloys are frequently used in aircraft and hulls, but serious corrosion and mechanical damage strongly affect the service life of these parts. The preparation of a dense, uniform, single-layer coating on an aluminum alloy can substantially improve the corrosion resistance of the alloy matrix, enhance the mechanical properties of the surface, and reduce the occurrence of accidents and property losses. This resolves the performance bottleneck of PEO coatings caused by the porous layer and broadens the market prospects for aluminum alloys.

## 2. Materials and Methods

### 2.1. PEO Preparation

We used industrial pure 1060 aluminum plate, provided by the Aluminum Corporation of China, as a substrate material for PEO preparation (see Table 1 for the aluminum composition). We used an Al plate (50 mm × 50 mm × 2 mm) and a graphite plate as the anode and cathode, respectively. We prepared aqueous electrolyte with distilled water containing 7 g/L Na_2_SiO_3_, 5 g/L NaH_2_PO_4_ and 3 g/L H_2_C_2_O_4_. Table 2 lists and describes the reagents involved in the experiment. We carried out the PEO process using a 50 kW self-made pulsed DC power source. We set the asymmetrical bipolar pulsed supply to a frequency of 200 Hz, and the duty ratios for the anode and cathode were 50% and 30%, respectively. We applied an anodic unipolar current pulse until we obtained a voltage of 480 V, and thereafter we used a bipolar current pulse. We prepared the coatings at a current density of 2 A/dm^2^ for 360 min and captured the discharge characteristics using a Canon DS126291 video camera during the PEO process.

### 2.2. Characterization and Analysis

We analyzed the microstructural characteristics and elemental composition of the samples using a scanning electron microscope (SEM, PHILIPS, XL–30FEG) equipped with an X-ray energy-dispersive spectrometer (EDS, Oxford, UK). We determined the phase constituents of the coatings using an X-ray diffractometer (XRD, PHILIPS, PW1700).

We performed electrochemical tests using a Zahner Zennium electrochemical workstation, to evaluate the corrosion resistance of the samples. We adopted a three-electrode system, using the sample as the working electrode, the saturated calomel as the reference electrode, and the platinum plate as the auxiliary electrode. We carried out all measurements at 30 ± 1 °C, in 3.5 wt.% NaCl solution. We selected the fixed mode for the potentiodynamic polarization curve test and obtained the anode and cathode curves at a sweep rate of 0.333 mV/s. After a 30 min immersion at open-circuit potential (OCP), we recorded the cathode curve from OCP to −300 MV relative to OCP and the anode curve from OCP until the current density reached 10 mA/cm^2^. We performed electrochemical impedance (EIS) tests over a frequency range of 100 KHz to 10 mHz using a 10 mV amplitude sinusoidal voltage. We analyzed the experimental data using the commercial software ZsimpWin and repeated each group of experiments at least three times.

To evaluate the PEO coatings’ adhesion strength, we performed three repetitions of an adhesion test based on the pull-off technique (according to ASTM D4541-02) and calculated the mean of the measurements. To perform this test, we prepared a pull stub with a diameter of 30 mm and stuck it to the coating specimens using an E7 structural adhesive, mainly composed of epoxy resin. Epoxy, as a polar resin, forms many hydrogen and van der Waals bonds with aluminum oxide [26]. We maintained the lateral displacement rate of the pull-out test beam at 0.5mm/min under a load of 10–20 N at 22 °C. We used a Japanese FM-700 hardness tester to test the hardness of the PEO coatings after randomly selecting five evenly distributed points on the sample surface. The load was 100 g and the holding time was 10 s. We made 5 indentations on each sample and calculated the average value.

We calculated the porosity of the coatings, by selecting 6 micro-areas on the cross-sectional SEM image of the PEO coating under a constant magnification, using the image analysis software image pro ^®^ Plus version 6.0 (for Windows™) developed by Media Cybernetics (Media Cybernetics, Company, Silver Spring, MD, USA). We calculated the average porosity value of the 6 areas, to determine the porosity of the coating.

## 3. Results and Discussion

### 3.1. Voltage–Time Curves and Discharge Evolution of Aluminum

Figure 1 and Figure 2 show the voltage–time curves and the evolution of the discharge during the PEO process, respectively. Figure 1 depicts the five stages of the PEO process according to the characteristics of the voltage variation over time, and Figure 2 illustrates the corresponding features of the discharge at each stage.

Stage I constitutes the initial stage of the PEO process; i.e., anodic oxidation. As shown in Figure 1, the voltage increased linearly with time during this stage. Meanwhile, the images in Figure 2 depicting the oxidation in minutes 3–10 show an intense gas evolution and the fading of the metallic luster. The voltage at this stage had not reached the breakdown voltage of the insulating film, so we did not observe any plasma spark discharge on the surface of the pure aluminum sample, indicating that a plasma reaction had not occurred. Thereafter, the voltage increase rate decelerated, and it was almost invariant at the end of stage II (shown in Figure 1). As the applied voltage exceeded the breakdown voltage of the insulating film, plasma spark discharge occurred on the surface of the sample. At this stage, the plasma spark experienced a series of evolutions. At the beginning of stage II, a glow discharge appeared on the sample surface. The oxidation images for minutes 12–15 demonstrate that the discharge point was very small and moved very quickly during the PEO process. As the voltage continued to rise, the small white sparks from early in stage II were replaced by larger and slower orange sparks, as represented in the images for minutes 22–43 in Figure 2. The discharge during this period is called “microarc” discharge. We only applied an anodic unipolar current pulse during stages I and II, before applying a bipolar current pulse when the positive voltage value rose to 480 V. At this stage, the positive voltage was basically constant, and the negative voltage increased sharply (see Figure 1), which meant that the coating was growing rapidly. This stage was marked by the appearance of “soft” spark discharge, which occurred alongside the arc discharge. During this stage, orange and white sparks flickered continuously. As the PEO duration increased, the spark discharge points decreased, and the intensity weakened. Afterwards, the absolute value of the negative voltage decreased, and the size and intensity of the microarc and soft-spark discharge gradually decreased. Since the system did not use up 100% of the Al^3+^ dissolved in the substrate, to calculate the efficiency of the PEO process, we included the film formation (ɳAl_2_O_3_), dissolved aluminum (ɳAl), and oxygen precipitation (ɳO_2_) efficiency. The expression is as follows:ɳAl_2_O_3_ + ɳAl + ɳO_2_ =100%

With the prolongation of the microarc oxidation duration, the concentration of Al^3+^ in the electrolyte rose, increasing the electrolyte conductivity [27]. At this point, the plasma spark discharge could be induced by a lower voltage; thus, the absolute value of the cathodic voltage decreased during stage IV. As shown in the images for minutes 140–290 min Figure 2, the microarc and soft-spark discharge weakened at the periphery of the specimen and became hardly visible, before increasing their area by spreading towards the center of the specimen [15]. As this conversion process concluded, slow-moving "chain-like” discharge appeared and became gradually more visible in stage V. The shape and position of this “chain-like” discharge were constantly changing, as shown in the images for minutes 340–360 in Figure 2. This may have been due to a gradual reduction in microdefects and the fact that plasma spark discharge always occurs at the weak points of a coating. Individual weak points are connected in series, to form “chain-like” discharge. When the blocking capacity of the coating at the point where “chain-like” discharge occurs exceeds that of the surrounding area, the “chain-like” discharge spontaneously changes its shape and moves to other weak points in the PEO coating. In stage V, the voltage remained almost constant, as shown in Figure 1, which indicates that the thickness of the coating did not increase rapidly.

### 3.2. Effect of Discharge Evolution on Microstructure of PEO Coating at Different Stages

The first generation of PEO, i.e., “arc-term PEO”, involved DC or pulsed DC regimes and formed a two-layer coating in silicate electrolyte, with an outer layer based on amorphous silica and an inner layer based on alumina [28]. By considering the microstructure of the PEO coating, Hussein et al. classified the discharge observed during the PEO treatment of an pure aluminum (1100 alloy) at a pulsed DC power mode in silicate electrolyte into three types: that which originated from the top coating or the gases attached to the coating surface (type A), from near the substrate/coating interface (type B), or from within the pores and cracks of the upper coating (type C). The coating material produced by type-B discharge is mainly composed of base metal oxide, while type A and C produce a material with more components from the electrolyte. The materials formed by different types of discharge on the coating surface display different morphologies; namely, pancake and nodular structures, respectively [29].

The second generation of PEO, i.e., “soft-sparking PEO”, was realized under specific AC or bipolar conditions [30,31,32,33]. The coatings usually comprised a highly porous outer layer, a relatively thick and dense intermediate layer, and a thin barrier layer [15]. Based on Hussein’s study, and considering the three-layer microstructure of these coatings, Y. L. Cheng et al. categorized the discharge observed during the soft-sparking PEO process into five types: types A, B, and C correspond to the abovementioned categorizations; type-D discharge occurs in the macropores near the inner and outer interfaces and increases the thickness of the inner layer and the merging of silicon species; and type-E discharge is relatively strong and may form pancake structures that are confined to the outer coating [34].

In this study, we propose a third generation of PEO, i.e., “chain-like discharge PEO”, by presenting novel discharge states that further affect the composition and microstructure of the PEO coating. Figure 3 depicts the surface morphology of the PEO coatings at different stages. Figure 4 presents schematic diagrams of the discharge type transitions according to the evolution of the plasma spark discharge state and the coating morphology at different stages of the PEO process.

In stage I, a thin oxide film formed on the sample surface, as evidenced by the scratches in the polished pattern (Figure 3a). The enlarged view (framed by the yellow line, Figure 3b) shows that the coating surface in stage II contained many through pores with a pancake microstructure. In addition, the occurrence of oxidation reactions at the solution–substrate interface caused the continuous release of gas from the point at which the coating surface came into contact with the electrolyte. Therefore, plasma spark discharge occurred at the gas–electrolyte (type A) and matrix–electrolyte (type B) interfaces. Figure 5 and Figure 6 depict the cross-sectional morphology and composition of the coating at different stages, further elucidating the influence of the change in discharge type on the microstructure and composition of the coating.

The films obtained in the anodic oxidation stage before plasma spark discharge displayed a stacking structure (Figure 5a). After breakdown, some discharge channels (Figure 5b) passed through the coating, owing to type-B discharge during stage II (through the pores shown in Figure 3b), which effectively dissolved the substrate and melted the oxide [28]. The existence of Si (shown in Figure 6a) on the surface of the coating indicated the occurrence of discharge types A and C. The XRD patterns of the PEO coatings at different stages (presented in Figure 7) showed that type-B discharge resulted in the formation of amorphous Al_2_O_3_, due to the rapid cooling process once the molten oxide made contact with the electrolyte. Alongside the main type-B discharge, types A and C occurred at a weaker intensity in stage II.

Figure 3c shows that the coating surface in stage III was rough, with many nodules and pores. Soft sparks appeared on the coupon surface after the application of a bipolar pulse current, resulting in the coexistence of white and orange sparks (Figure 2). S. P. Sah et al. demonstrated that cathode discharge occurs at the point where anode breakdown has just occurred, and that the coating produced by cathode discharge is much denser than that produced by anode discharge. Consequently, the inner layer (Figure 5c, yellow dotted line) at the coating–substrate interface is fabricated rapidly, because the increased resistance of the coatings after cathodic discharge changes the location of the next anodic discharge to another weak point [35]. After stage III, four interfaces were formed between the substrate and the coating layer: gas–solution, substrate–solution, loose porous external layer–barrier layer, and barrier layer–substrate. As with type-A, -B, and -C discharge, type-D discharge is initiated by the inner/outer layer microstructure (Figure 4b). Type-D discharge occurs at the outer–inner interface and promotes the growth of the dense inner layer and the incorporation of silicon species [34]. During stage III, the coatings mainly grew inward, establishing a relatively continuous barrier layer, and the composition and material phases of the coatings were almost the same as during stage II (Figure 6 and Figure 7). Therefore, type-B and -D discharge controlled stage III.

Stage IV is also called the “transformation” stage. Due to the “edge” effect of plasma spark discharge (see Figure 2), the first transformation coating was thicker, and the later was thinner (Figure 5d1). During this stage, type-B and -D discharge cooperated with each other, to generate matrix oxides and increase the thickness of the coating. Early in stage IV, the discharge type in the thinner area of the coating was the same as in stage III, while the dominance of strong type-B plasma spark discharge in the thicker part of the coating was replaced by type-E plasma spark discharge occurring at the loose porous external layer–solution interface (Figure 4c). Alongside nodules, fine particles appeared on the coating surface during stage IV (Figure 3d). Accordingly, Figure 6d exhibits the increased amount of Si on the thicker coating surface. Therefore, type-A, -C, and -D discharge dominated in stage IV, resulting in the deposition of solution components and an increased thickness of the dense layer. Moreover, careful observation of Figure 7d reveals the increased diffraction peak intensity of γ-Al_2_O_3_ compared to Figure 7c. This could be ascribed to the increased compactness of the coating, which more effectively retained a high temperature, leading to the transformation of amorphous Al_2_O_3_ into γ-Al_2_O_3_. Over time, the entire coating became uniform, displaying a thick dense layer towards the end of stage IV (Figure 5d2).

In stage V, the type-E discharge melted and condensed the loose outer layer, and type-A and -C promoted oxidation reactions within the solution. With the extension of the PEO duration, many nodules on the surface of the sample were replaced by a pancake microstructure (with pores as shown in Figure 3e) sealed by oxide, which we ascribed to the impact of the interfacial discharge [34]. The type-E discharge occurring at the loose porous external layer–solution interface also produced a pancake microstructure on the surface of the coating, but this was unlikely to cause defects similar to the through pores in the coating generated by type-B discharge. In addition, we observed further augmentation in the Si content on the outer coating and an increase in the diffraction peak intensity of α-Al_2_O_3_ and mullite (Figure 7e). This took place because of the improved density of the coating, which preserved the high temperature generated by the discharge and thus promoted the transformation of γ-Al_2_O_3_ into α-Al_2_O_3_. The formation of “chain-like” discharge (shown in Figure 2) was essentially the result of a series of processes that were dominated by type-A, -C, and -E discharge. Thus, the discharge type affected the microstructure and material phases of the coating, and a change in the microstructure altered the dominant discharge type. In other words, the “chain-like” discharge and dense microstructure of PEO coatings are reciprocal. Finally, the “chain-like” discharge substantially improved the compactness of the porous outer layer; Figure 5e presents the single dense layer with few defects obtained in stage V.

### 3.3. Impact of Microstructure on the Corrosion Resistance of PEO Coating at Different Stages

The above discussion elucidates the significant influence of the discharge evolution throughout the different stages on the morphology, microstructure, and composition of the PEO coating, which further affects its protective performance. We carried out EIS to evaluate the corrosion behavior of the coatings at different stages. Figure 8a,b depicts the Bode plots and simplified equivalent circuits.

We identified two time constants, at low and medium frequencies, from the Bode plot of the coating in stage II and used mode A to fit the EIS data. The PEO coating capacitance and resistance (R_PEO_ and Q_PEO_) represent the response of the entire PEO coating, while the double-layer capacitance and charge-transfer resistance (Q_dl_ and R_ct_) indicate the reaction between the corrosive agent and the substrate at a low frequency. We identified three time constants from the Bode plots of the coating in stage III and IV and simulated the experimental data using mode B. The fit parameters of (R_out_, Q_out_) and (R_in_, Q_in_) correspond to the responses of the porous outer and dense inner layers, respectively. Nevertheless, the impedance value |Z| of the coating in stage IV was higher than that in stage III. For the coating in stage V, we observed two time constants (Figure 8a). Unlike the Bode plot of stage II, the time constants appeared at high and medium frequencies. The phase angle was greater than 80 in a wide frequency range, which implies that the coating functioned as an insulation layer with effective capacitance characteristics [36]. Figure 8b shows the corresponding equivalent circuits (mode C), and Table 3 lists the fit results.

The R_in_ of the coating obtained after the appearance of “chain-like” discharge was almost three orders of magnitude higher than that obtained under other discharge conditions. Thus, the “chain-like” discharge significantly reduced the defects in the inner layer, which efficiently blocked the permeation of the corrosive agent.

Figure 9 illustrates the potentiodynamic polarization curves of pure Al with and without a PEO coating, which we analyzed to assess the corrosion resistance of the PEO coating at different stages. Table 4 lists the corresponding electrochemical parameters obtained by the Tafel extrapolation method. According to the polarization curve, the anodic and cathodic reactions of the pure Al with a PEO coating achieved an obvious inhibition, with the prolongation of the PEO treatment duration. The self-corrosion current density (*i_corr_*_)_ of pure Al is 1.629 × 10^−4^ A∙cm^−2^, which was decreased by an order of magnitude due to the anodic oxidation film formed on the Al in stage I. Although the porosity of the coating in stage I was lower than in stage II (as shown in Figure 10), the *i_corr_* of the coating in stage II decreased by an order of magnitude compared with the anodic oxidation coating in stage I. This was because the anodic oxidation film comprised a stacking structure, which allowed the corrosive medium to quickly reach the matrix through the gaps between layers, resulting in metal corrosion. However, the PEO coating in stage II contained many micropores. The corrosion medium had to bypass these pores, which prolonged the time it took to arrive at the matrix and reduced the contact area between the corrosion medium and the matrix, thus diminishing the corrosion rate. The *i_corr_* of the coating in stage III was similar to that in stage II. Although the porosity of the coating was the highest in stage III, a discontinuous barrier layer formed between the substrate and the outer layer early in this stage, owing to the application of a negative pulse. This further hindered the transmission of the corrosive medium compared to the coating in stage II. Consequently, the *i_corr_* decreased slightly. However, the *i_corr_* of the coating in stage III decreased by two orders of magnitude compared with the Al matrix, indicating that the barrier layer was very important for improving the corrosion resistance of the PEO coating. The cross-sectional morphology showed that the thickness of the coating in stage IV increased substantially, and the dense layer was thicker than in the previous stage. However, the porosity did not decrease substantially, because the loose-layer-to-dense-layer ratio was still very high at this stage, and the coating contained many micropores and cracks. Figure 9 shows that the anode and cathode curves of the coating in stage IV were considerably suppressed, and the *i_corr_* decreased by an order of magnitude compared with stage III. This indicates that the relatively dense inner layer expanded the path of the corrosion medium as it penetrated through the coating. However, with the increase in the electrode potential, the corrosion current density of the coating in stage IV approached that of the coating in stage III, which illustrates that the contact area between the corrosion medium and the substrate was similar. Therefore, the density of the coating is the key to reducing the corrosion rate. The porosity of the coating with a single dense layer in stage V reached as low as 2.8%, and the *i_corr_* decreased by four orders of magnitude compared with the matrix. With the increase in the anode potential, the final corrosion current density was less than 10^−6^ A∙cm^−2^, which indicates that the dense microstructure effectively reduced the contact area between the corrosion medium and the matrix. With the prolongation of the PEO treatment time, the *E_corr_* of the coating gradually increased from −0.457 V to −0.335 V, which means that the corrosion resistance of the coating with a single dense layer was considerably improved.

### 3.4. Effect of Microstructure and Phase Composition on the Mechanical Properties of PEO Coating at Different Stages

Figure 10 shows the adhesion test results. The bonding strength of a coating depends on three factors: First, the PEO coating forms a uniform structure which provides a chemical bond between the coating and the epoxy resin. Second, cracks in the PEO coating constitute bonding failure points [26]. Third, the porosity of the exterior structure of a PEO coating increases the size of the effective area, enhancing the adhesive strength between the epoxy resin and the coating [37]. In other words, the presence of pores and defects in the outer layer of the PEO coating creates a surface appropriate for the epoxy coating. Therefore, the adhesion of the coating in stage II is stronger than in the anodic oxidation stage (stage I). The specimens prepared using the unipolar-then-bipolar pulse system (DC + AC) in stage III demonstrated greater adhesion because of the preliminary barrier layer, which strengthened the epoxy coating’s adhesion through the porous interface of the PEO coating. Compared with the coating in stage III, the adhesion of the coating in stage IV was slightly improved. This was due to the non-uniformity of the coating, according to the cross-sectional scanning photos (see Figure 5d1). The coating in the transformed area was thicker, while the coating in the non-transformed area was thinner. Failure first occurs in thinner areas. In addition to the barrier layer, the number and size of the pores affected the adhesion strength of the top layer. A surface with proper uniformity facilitates mechanical bonding and demonstrates more extensive and effective contact with the epoxy coating [38]. The coating in stage V displayed the greatest adhesion strength of the samples, probably because of its finer porosity and more uniform structure.

The XRD spectra showed that the phase composition of the PEO coating varied between the different stages, which further affected its hardness. Figure 11 illustrates the Vickers hardness values of the PEO coating at different stages. The hardness of the coating in stage I was about 400HV, and the hardness of the film in stages II and III was between 700 and 850 HV. This was because the film formed in stage I was mainly composed of amorphous Al_2_O_3_ (Figure 7). After the high temperature effect of plasma spark discharge, the amorphous Al_2_O_3_ became γ-Al_2_O_3_, which hardened the coating. Figure 11 demonstrates that the hardness of the coating reached a turning point during stage III, increasing substantially thereafter. The hardness of α-Al_2_O_3_ is greater than that of γ-Al_2_O_3_. The thickness and compactness of the coating increased under the bipolar pulse mode (shown in Figure 5d), which promoted the transformation from γ-Al_2_O_3_ to α-Al_2_O_3_. The hardness of the coating in stage V reached as high as 2000HV, due to the single dense layer. Therefore, the microstructure of the coating affected the phase composition of the PEO coating at different stages, which further impacted its hardness.

The compact microstructure of PEO coatings obviously plays a key role in improving their corrosion resistance and mechanical properties, and the plasma spark discharge state is a critical determinant of the microstructure of aluminum coatings. Therefore, adjusting the plasma spark discharge state to obtain a single dense layer is the fundamental means of improving the performance of a coating.

## 4. Conclusions

In this study, the evolution of discharges during the PEO process has been described, and novel “chain-like” discharges are revealed. We demonstrated that this novel “chain-like” discharge can yield a uniform and compact single-layer coating. The achievement of a single-dense layer coating by adjusting the plasma spark discharge, as demonstrated in this study, offers a new technique for the densification of PEO coatings on other alloys. This solves the bottleneck problem that has plagued researchers for a long time, whereby the ratio of loose layer to dense layer is high and affects the performance of the PEO coatings. It also expands the application prospects for aluminum alloys in national defense equipment and delivery vehicles. Our conclusions are as follows:

(1)According to the voltage–time curve and the evolution of the plasma spark discharge state under the conditions of unipolar + bipolar (DC + AC) pulses, the PEO process comprises five stages: the anodic oxidation stage (I), the microarc discharge stage (II), the soft-spark stage (III), the transformation stage (IV), and the “chain-like” discharge stage (V).(2)The plasma spark discharge state affects the microstructure and phase composition of the PEO coating. The morphology of the PEO coating changes across the entire PEO process: stage I comprises a single layer of stacked insulating film; a layer of molten oxide coating forms in stage II; during stage III, a barrier layer forms, and the coating presents a two-layer structure; the dense layer thickens considerably after the transformation stage (IV); and the “chain-like” discharge stage (V) produces a single dense layer with few defects. The diffraction peak intensities of the γ-Al_2_O_3_, α-Al_2_O_3_, and 3Al_2_O_3_ · 2SiO_2_ phases increases as the PEO treatment duration increases.(3)The type of plasma spark discharge affects the coating formation mechanism: type-A discharge represents gas discharge, which encourages the solution components to participate in oxidation reactions; type-B discharge demonstrates a high intensity, which effectively dissolves the matrix and releases a large amount of gas; type-C and -D discharge reduce the defects in the barrier layer and dense inner layer; and type-E discharge melts and condenses the loose external layer. Unlike traditional plasma spark discharge, “chain-like” discharge mainly cooperates with discharge types A, C, D, and E. The “chain-like” discharge moves continuously over the whole sample, which effectively reduces the defects and increases the uniformity of the coating, substantially improving the corrosion resistance and mechanical properties of the PEO coating.

## Figures and Tables

**Figure 1 materials-15-04635-f001:**
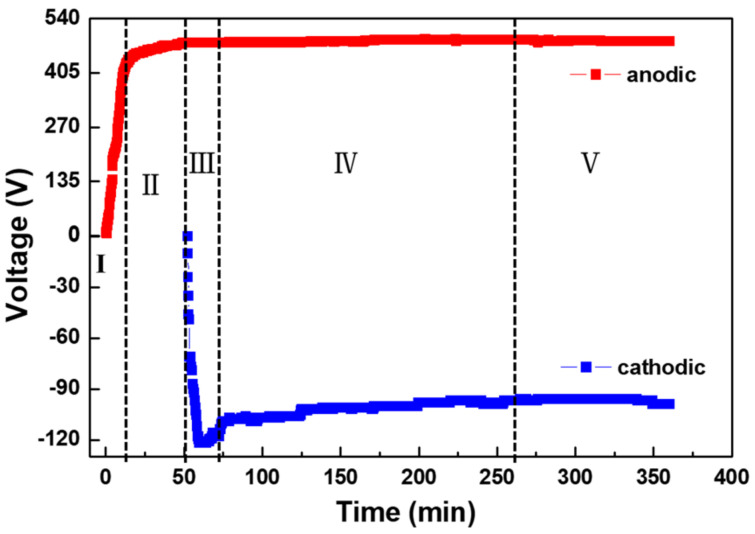
Voltage−time curves for the PEO process.

**Figure 2 materials-15-04635-f002:**
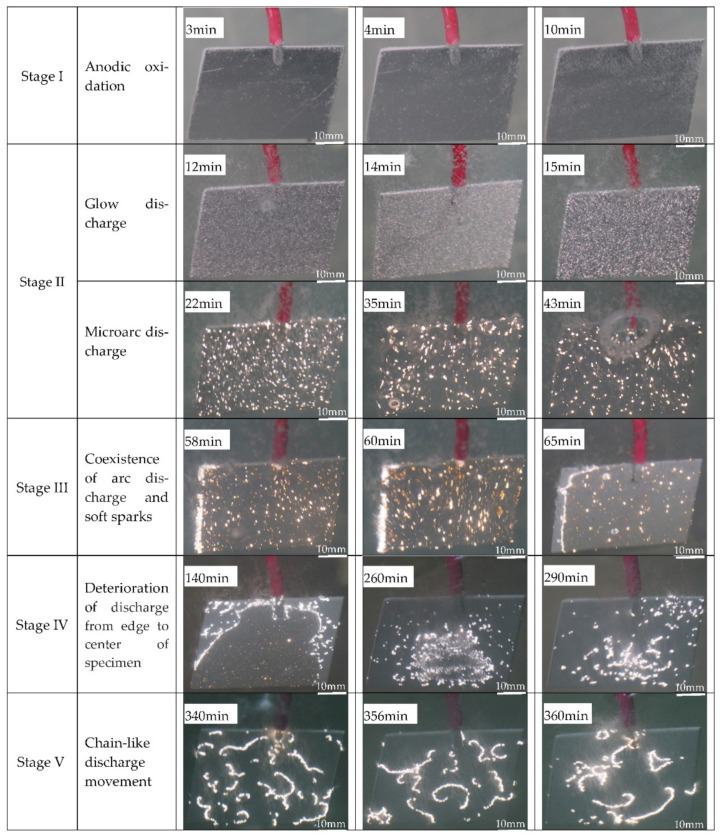
Evolution of microdischarge throughout the entire PEO process.

**Figure 3 materials-15-04635-f003:**
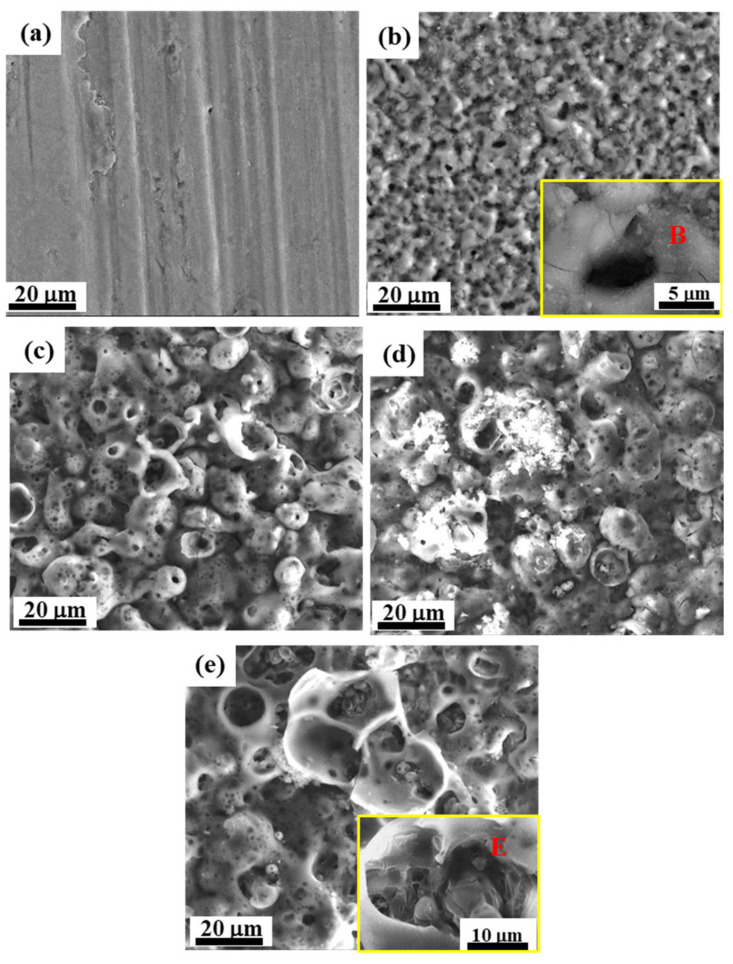
Surface morphology of PEO coating at different stages: (**a**) stage I; (**b**) stage II (Type B discharge framed by the yellow line); (**c**) stage III; (**d**) stage IV; (**e**) stage V (Type E discharge framed by the yellow line).

**Figure 4 materials-15-04635-f004:**
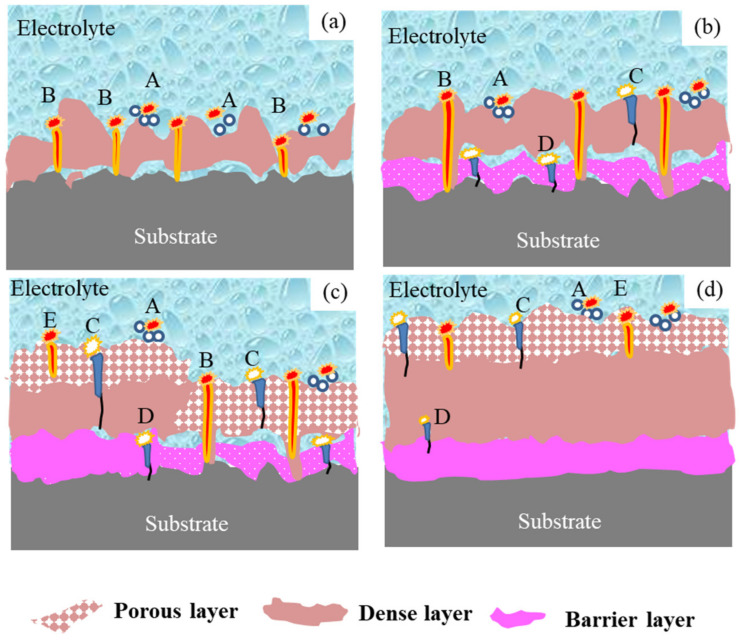
Schematic diagram of discharge type transitions forming a dense and uniform single layer: (**a**) stage II; (**b**) stage III; (**c**) early period of stage IV; (**d**) later period of stage IV.

**Figure 5 materials-15-04635-f005:**
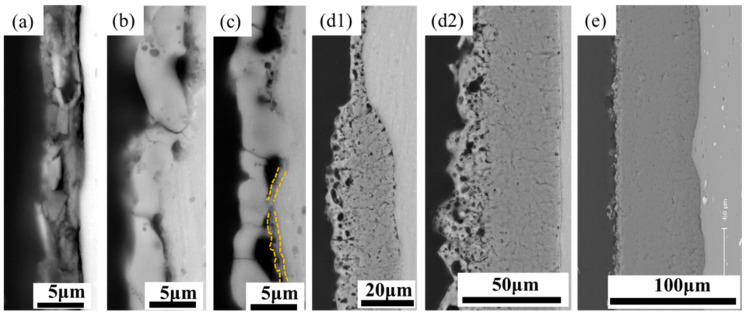
Cross-sectional micrographs of PEO coatings at different stages: (**a**) stage I; (**b**) stage II; (**c**) stage III; (**d1**) early period of stage IV; (**d2**) later period of stage IV; (**e**) stage V.

**Figure 6 materials-15-04635-f006:**
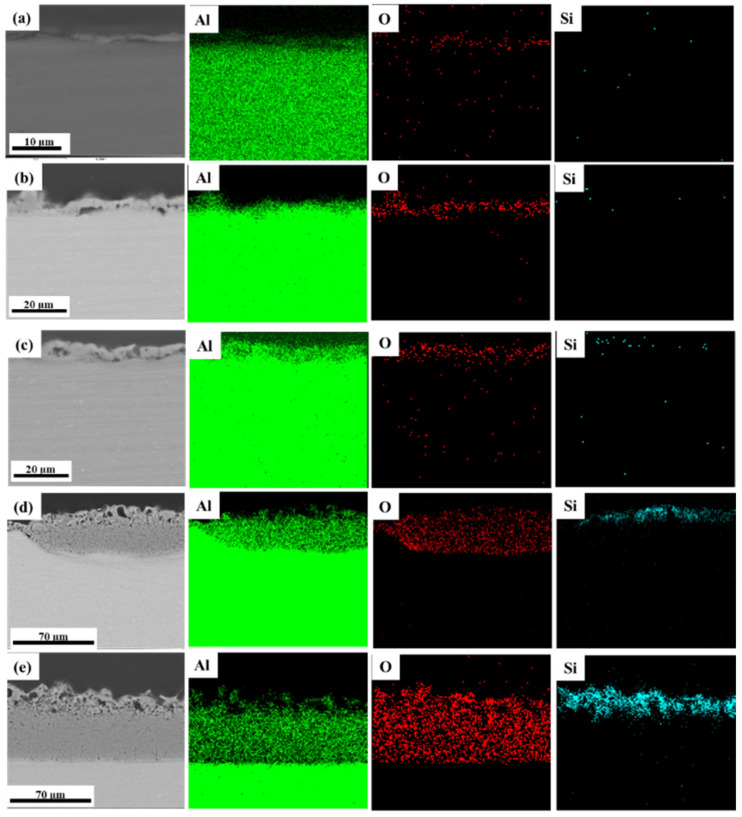
EDS mapping images of PEO coatings at different stages: (**a**) stage I; (**b**) stage II; (**c**) stage III; (**d**) early period of stage IV; (**e**) later period of stage IV.

**Figure 7 materials-15-04635-f007:**
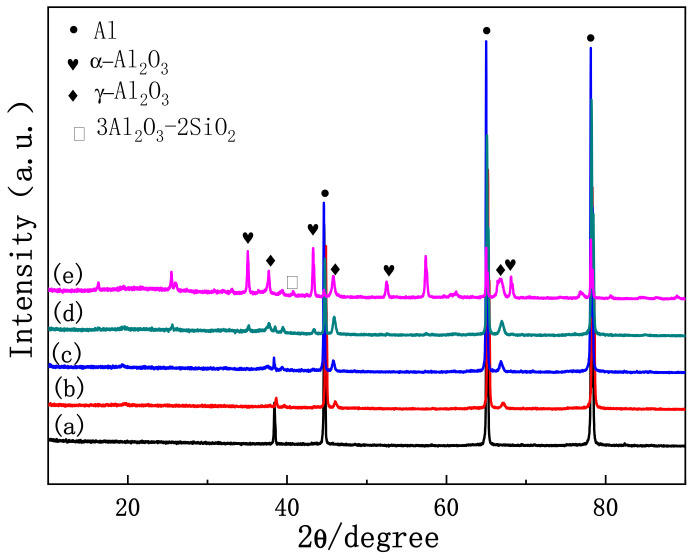
XRD patterns of PEO coatings at different stages: (**a**) stage I; (**b**) stage II; (**c**) stage III; (**d**) stage IV; (**e**) stage V.

**Figure 8 materials-15-04635-f008:**
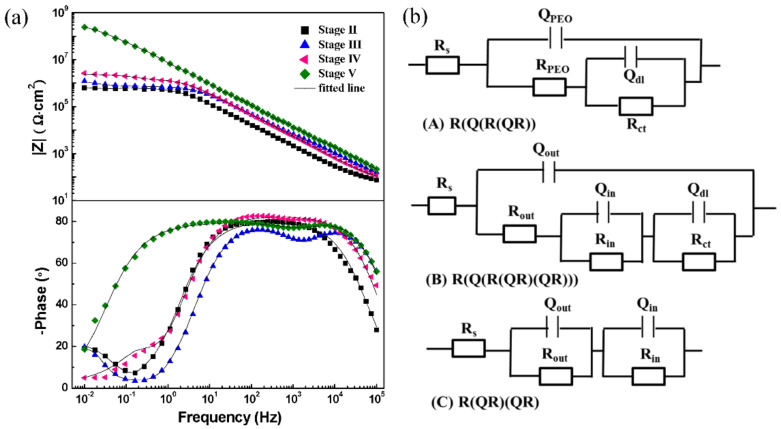
(**a**) Bode plots and (**b**) equivalent circuits of PEO coatings at different stages.

**Figure 9 materials-15-04635-f009:**
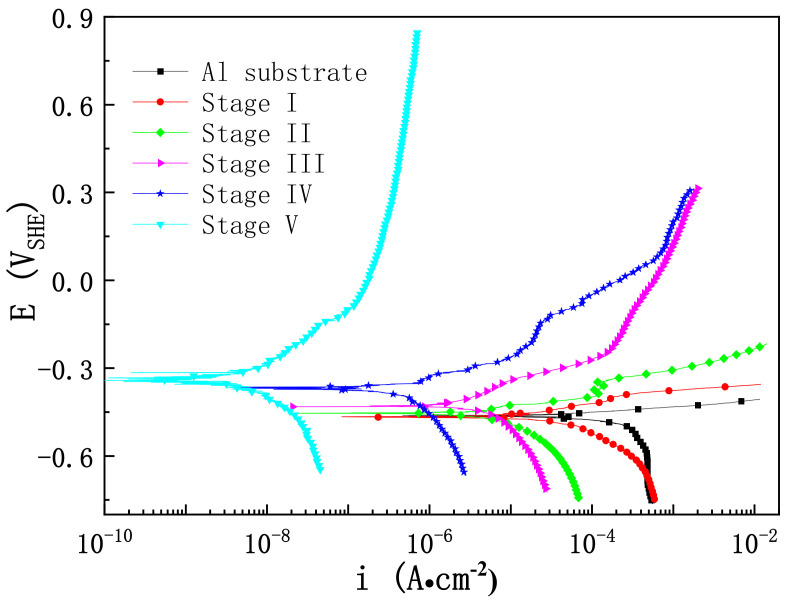
Potentiodynamic polarization curves for pure Al with and without PEO coating at different stages in 3.5 wt.% NaCl solution.

**Figure 10 materials-15-04635-f010:**
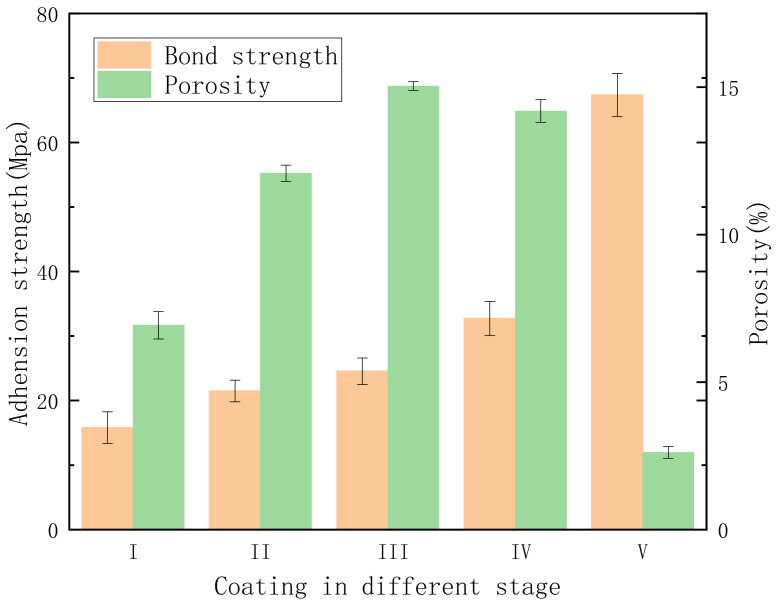
Adhesion and porosity of PEO coatings at different stages.

**Figure 11 materials-15-04635-f011:**
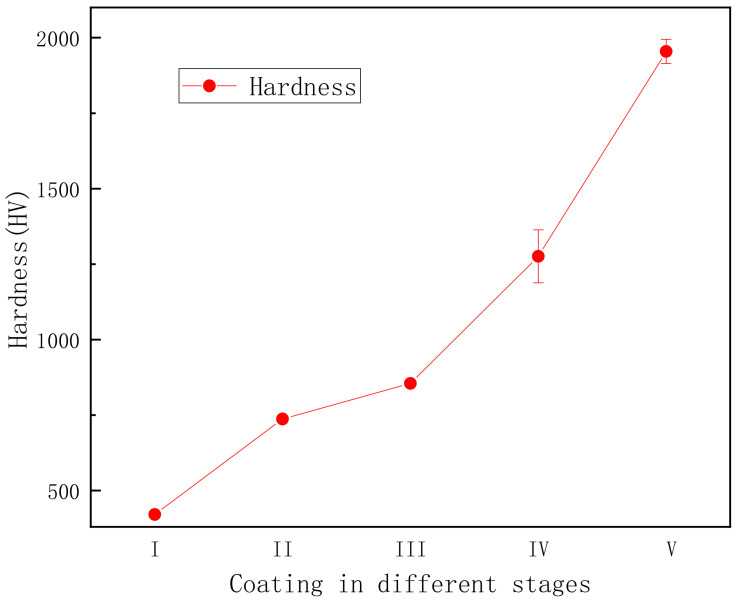
Vickers hardness variation of the PEO coating at different stages.

**Table 1 materials-15-04635-t001:** Composition of industrial pure 1060 aluminum (wt.%).

Element	Al	Si	Cu	Mg	Zn	Mn	Ti	V	Fe
Percentage	99.60	0.25	0.05	0.03	0.05	0.03	0.03	0.05	0.350

**Table 2 materials-15-04635-t002:** Experimental reagents.

Reagent Name	Molecular Formula	Reagent Purity	Manufacturer
Sodium dihydrogen Phosphate dodecahydrate	NaH_2_PO_4_·12H_2_O	Analytically pure	Sinopharm Chemical ReagentCo., Ltd. Shenyang, China
Oxalic acid dihydrate	H_2_C_2_O_4_·2H_2_O	Analytically pure	Sinopharm Chemical ReagentCo., Ltd. Shenyang, China
Sodium silicate nonahydrate	Na_2_SiO_3_·9H_2_O	Analytically pure	Sinopharm Chemical ReagentCo., Ltd. Shenyang, China

**Table 3 materials-15-04635-t003:** Equivalent circuit data for PEO coatings at various stages.

Stage	R_out_ (Ω·cm^2^)	Q_out_ (S^n^·Ω^−1^·cm^−2^)	n	R_in_ (Ω·cm^2^)	Q_in_ (S^n^·Ω^−1^·cm^−2^)	n	R_ct_ (Ω·cm^2^)	Q_dl_ (S^n^·Ω^−1^·cm^−2^)	n	χ^2^
II				6.74 × 10^5^	1.20 × 10^−7^	0.89	7.05 × 10^5^	3.21 × 10^−5^	0.80	1.42 × 10^−3^
III	1.27 × 10^4^	3.74 × 10^−8^	0.92	7.09 × 10^5^	2.77 × 10^−8^	0.88	7.17 × 10^5^	1.73 × 10^−5^	0.93	1.30 × 10^−3^
IV	3.56 × 10^5^	2.29 × 10^−5^	0.88	7.17 × 10^5^	1.54 × 10^−6^	0.95	1.50 × 10^6^	5.75 × 10^−8^	0.94	5.68 × 10^−4^
V	1.98 × 10^8^	2.42 × 10^−8^	0.85	1.56 × 10^9^	4.96 × 10^−8^	0.87	-	-	-	8.32 × 10^−4^

**Table 4 materials-15-04635-t004:** Electrochemical parameters of PEO coating on Al at different stages.

Stage	*E*_corr_ (V _SHE_)	*i*_corr_ (A∙cm^−2^)
Al	−0.457 ± 0.013	1.629 ± 3.804 × 10^−4^
Stage I	−0.460 ± 0.011	2.886 ± 2.325 × 10^−5^
Stage II	−0.438 ± 0.023	5.282 ± 1.384× 10^−6^
Stage III	−0.426 ± 0.018	1.896 ± 1.278 × 10^−6^
Stage IV	−0.372 ± 0.034	4.714 ± 4.143 × 10^−7^
Stage V	−0.335 ± 0.027	8.564 ± 5.763 × 10^−9^

## Data Availability

The data used to support the findings of this study are available from the corresponding author upon request.

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
