# Peer review of "Single Dense Layer of PEO Coating on Aluminum Fabricated by “Chain-like” Discharges"

_materials, 2022, doi:10.3390/ma15134635_

Round 1
Reviewer 1 Report
General remarks
1. ABSTRACT is very short, below 100 words. It does not show the experimental problem (background), or used methods (even a base material). A reader may not be interested in reading the whole paper after reading such a description. The abstract must be rewritten and enlarged.
2. INTRODUCTION is very comprehensive and does not correspond to a classical form of the state-of-the-art. It cannot be accepted in its present form. There is nothing about a mechanism of PEO as related to different parameters. Then, all references should demonstrate details: what metals, what process parameters, and what results; without that, a reader I not be convinced that this research is different from the others, in which?
3. The authors claim that it is a dense layer, but such compactness has not been tested by, e.g., hardness or scratch tests.
4. EIS test is not enough for establishing the improvement of corrosion resistance. Make potentiodynamic curves for all specimens finding the corrosion potential and current density.
5. Finally, I cannot understand the idea of this research reading conclusions: “The present study revealed that the evolution of discharges i.e. discharge types in PEO process has considerable impact on the microstructure of the coating. The novel “chain-like” discharges can yield uniform and compact microstructure with thin porous layer, which could be a new way to improve anti-corrosion property of PEO coating”. My most important remark is that the authors have not applied any independent variable setting only single values of voltage, current, and oxidation time. Justify that such behavior will be typical for all other process parameters, what is a novelty of such an approach, and what is the difference between the previous and present results. That following, the paper needs a substantial improvement in both additional tests and description.
Specific remarks
6. Page 1: “Because the “chain-like” discharges are mainly dominated by A type, C type and E type discharges which can further reduce the microdefects of the coating and increase the electrolyte deposition.” The sentence is unclear. What types of discharges further reduce, C and E? Why further? What microdefects? Why do they increase the deposition of electrolyte, rather than a coating?
7. Page 1: “The “chain-like” discharges moves continuously on the surface of the sample and finally effectively improves compactness of porous layer”. The same as above is unclear. Why do discharges move over the surface? Why do they improve compactness?
8. Page 1: “Porous layer with a large number of through flaws”. What do the authors consider as through flaws? What is the real porosity, or roughness of so far obtained layers?
9. Page 1: “Therefore, reducing the porous layer and increasing the proportion of dense layer”. Reducing of what, the porosity of thickness? The proportion of what to what?
10. Page 1: “An attempt is made to divide five stages of the PEO process”. Why five stages? Besides, the division of the PEO process into five stages can be a result of an analysis and not an attempt.
THE MATERIAL AND METHODS section is well-prepared, as a rule. Specific remarks:
11. Page 2: Give a delivering/producing company for Al plate.
12. Page 2: Give producers of all chemicals, and their chemical purities.
13. Page 2: Give a type and a manufacturing company for a power supply.
14. Page 2: “duty ratios obtained for the anode and cathode were 50% and 30%”. What does it mean, duty ratios?
15. Page 2: „2 A/dm2”. Such current density was constant during the whole PEO process?
16. Page 2: Give details for the EIS device.
RESULTS AND DISCUSSION
17. Fig. 1 shows the anodic and cathodic runs. Please come back to the methodology and improve the oxidation process description showing what they mean, anodic and cathodic curves, between what and what, etc.
18. Page 2: I do not agree that the oxidation process has five stages. At first, an appearance of the third stage is possible simply only as a consequence of bipolar current. Next, even taking a look at both anodic and cathodic curves, I see no difference between regions IV and V. Such difference is only as surfaces are examined.
19. Page 2: what does it mean “fading of metallic luster”?
20. Page 2: “with slower in movement”. Something is wrong, what is slower and what movement?
21. Page 4: “cathodic discharges changes the next anodic discharges to another weak point”. Fully unclear, anodic discharges may change cathodic discharges, by what mechanism, are they dependent? And to what weak point, what do you mean?
22. Page 5: “series connection”; improper phrase, I suppose; series of connections? And connections between what and what?
23. Figure 2: a reader is not convinced whether the observed effects are not incidental. Please give more photos, at different images, for each stage.
24. Next, the authors claim that they have obtained not a three-layer, but a single dense layer. Explain why both other layers vanished.
25. Grammar errors: the improper and proper (after semicolon) phrases are given:
· Page 1: the “chain-like” discharges is mainly dominated; … discharges … are …
· Page 1: discharges moves; discharges move.
· Page 1. Porous layer with a large number of through flaws are very…; …layer … is …
· Page 1: An attempt is made to divide five stages of the PEO process; to divide PEO process into five stages.
· Page 1: corrosion resistance of coating in various stage were studied; … resistance … was …
· Page 3: is mainly attribute; is … attributed.
· Page 3: The second generation of PEO, i.e. “soft sparking PEO”, which realized under selected AC or bipolar PEO conditions; … generation … has been run … (`which` is unnecessary word).
· Page 4: cathodic discharges changes the next anodic discharges; …discharges … change …
Author Response
Dear reviewers:
Thank you for your comments concerning our manuscript entitled “Single dense layer of PEO coating on aluminium fabricated by “chain-like” discharges” (materials-1771898). Those comments are all valuable and very helpful for revising and improving our paper, as well as the important guiding significance to our researches. We have studied comments carefully and the following is a point-to-point response to your comments. Please see the attachemnt

Reviewer 2 Report
Review report on the topic ‘Single dense layer of PEO coating on aluminium fabricated by “chain-like” discharges’. Comments are listed below:
1. Strengthen the abstract section. Add the key conclusion of the works in the last two lines of the abstract section.
2. Discuss the motive behind the work. The clear application of the work should be discussed in the introduction section. From the introduction section application of the work is not clear.
3. There are numerous spelling and grammatical errors. Please revise the manuscript thoroughly. Sentences are also not complete.
4. The novelty of the work should also be discussed in a separate paragraph.
5. Try to make a bridge between current and previously published work and specify the gap area and objective of the work. Add the specific gap observed from the literature at the end of the introduction section. Refer to following:
https://doi.org/10.1007/s12540-020-00705-w; https://doi.org/10.1016/j.ceramint.2018.01.131;
6. Provide more detail about the experimental section.
7. The manuscript is written very poorly. A lot of English and grammatical errors in the manuscript. Please improve the quality of the writing.
8. Technical discussion is very poor. Improve it.
The manuscript is written well and can be accepted after following minor corrections.

Author Response
Dear reviewers:
Thank you for your comments concerning our manuscript entitled “Single dense layer of PEO coating on aluminium fabricated by “chain-like” discharges” (materials-1771898).Those comments are all valuable and very helpful for revising and improving our paper, as well as the important guiding significance to our researches. We have studied comments carefully and the following is a point-to-point response to your comments. Please see the attachment.

Reviewer 3 Report
The evolution of the plasma electrolytic oxidation (PEO) process is well known, and research in this area is constantly evolving. This process of protecting the surface of an aluminum part is considered one of the most energy efficient.
In this article, the authors highlighted the coating formation kinetics and evaluated the structure and properties obtained.
The procedure for implementing the PEO process, described in the article, is correct and achieves its purpose of highlighting the evolution of discharges in the PEO.
In principle, the bibliographic references are too few considering the numerous researches that have been carried out in this field. Let me give you just 2 examples, one of which is oriented in the same direction as this article.
V.S. Egorkin, S.V. Gnedenkov, S.L. Sinebryukhov, I.E. Vyaliy, A.S. Gnedenkov, R.G. Chizhikov, Increasing thickness and protective properties of PEO-coatings on aluminum alloy, Surface & Coatings Technology 334 (2018) 29-42
Dong-dong Wang, Xin-tong Liu, Ye-kang Wu, Hui-ping Han, Zhong Yang, Yu Su, Xu-zhen Zhang, Guo-rui Wu, De-jiu Shen, Evolution process of the plasma electrolytic oxidation (PEO) coating formed on aluminum in an alkaline sodium hexametaphosphate ((NaPO3)6) electrolyte, Journal of Alloys and Compounds 798 (2019) 129-143
Please also allow me to send you the following comments:
- Of course, researchers in the field should know what PEO means, but I think you need to specify, at least in summary, PEO (Plasma Electrolytic Oxidation)
- Commercial pure aluminum ???? you must specify exactly what aluminum you used;
- Corrosion property of the PEO coating was evaluated .... I could not identify the result of this evaluation;
- in figure 2 you only explain Fig.2a, I assume that the others (b ... d) correspond to stage III ... V?
- the paragraphs with The first generation of PEO ... and The second generation of PEO ... should have been included in the introductory chapter;
- the explanations in figure 3 must appear before the proper figure, also, not all the elements in figure 3 are explained;
- the conclusions do not add value to the article and need to be improved, at least with practical recommendations for the application of PEO.
In order to be interesting for researchers, you must also perform an evaluation of the mechanical properties.
Please highlight the novelty brought by your research.
Author Response
Dear reviewers:
Thank you for your comments concerning our manuscript entitled “Single dense layer of PEO coating on aluminium fabricated by “chain-like” discharges” (materials-1771898).Those comments are all valuable and very helpful for revising and improving our paper, as well as the important guiding significance to our researches. We have studied comments carefully and the following is a point-to-point response to your comments. please see the attachment

Round 2
Reviewer 1 Report
I appreciate a significant effect in improving both a research and a manuscript. That following, I am fully satisfied and have no further remarks.
Reviewer 2 Report
Accepted.
Reviewer 3 Report
The authors have substantially improved the quality of the article.